# A Survey of Energy-Efficient Communication Protocols with QoS Guarantees in Wireless Multimedia Sensor Networks

**DOI:** 10.3390/s19010199

**Published:** 2019-01-07

**Authors:** Shu Li, Jeong Geun Kim, Doo Hee Han, Kye San Lee

**Affiliations:** 1Advanced Wireless Communication Lab, Department of Electronics and Radio Engineering, Kyung Hee University, Yongin-si, Gyeonggi 17104, Korea; bluemindstorm@khu.ac.kr (S.L.); hdh9038@khu.ac.kr (D.H.H.); 2Multi-Service Networking Lab, Department of Electronics and Radio Engineering, Kyung Hee University, Yongin-si, Gyeonggi 17104, Korea; jg_kim@khu.ac.kr

**Keywords:** wireless multimedia sensor networks, energy-efficiency, wireless communication protocols, multipath routing, medium access control, quality of service

## Abstract

In recent years, wireless multimedia sensor networks (WMSNs) have emerged as a prominent technique for delivering multimedia information such as still images and videos. Being under the great spotlight of research communities, however, multimedia delivery over resource- constraint WMSNs poses great challenges, especially in terms of energy efficiency and quality-of-service (QoS) guarantees. In this paper, recent developments in techniques for designing highly energy-efficient and QoS-capable WMSNs are surveyed. We first study the unique characteristics and the relevantly imposed requirements of WMSNs. For each requirement we also summarize their existing solutions. Then we review recent research efforts on energy-efficient and QoS-aware communication protocols, including MAC protocols, with a focus on their prioritization and service differentiation mechanisms and disjoint multipath routing protocols.

## 1. Introduction

Wireless multimedia sensor networks (WMSNs) have become a promising research area in recent years. Unlike traditional wireless sensor networks (WSNs), which are typically deployed to collect scalar data (e.g., temperature and humidity), WMSNs are also able to collect multimedia data including audio, still image, video, and even live media streams with the addition of a microphone and a camera to the sensor nodes. With significant advances in embedded systems, image processing techniques, and communication technologies, WMSNs are anticipated to be used for a wide range of applications, such as security surveillance, real-time traffic and environmental monitoring, health care, and so on [1,2,3].

As a descendant of traditional WSNs, WMSNs inherit most of their advantages including deployment simplicity, scalability, resilience, and self-organizing capabilities. However, WMSNs are also confronted with many new challenges, one of which is the demand for highly energy-efficient and QoS-aware computation algorithms and communication protocols. Such requirements are crucial because WMSNs are by nature quite resource constrained in terms of energy, processing capacity, and communication bandwidths, while at the same time they also generate large volumes of multimedia data that need to be delivered over the shared wireless medium urgently and reliably. Consequently, most techniques originally developed for traditional WSNs cannot be directly used in WMSNs. Instead, new techniques ranging from the application layer down to the physical layer need to be designed and existing ones from traditional WSNs should get modified before they are applied to WMSNs. Fortunately over the years, the research community has done a lot of work, and there have been various surveys that sum up such previous efforts. Leading the way are the top-down surveys that cover from sensor hardware to every individual layer in the network model, as well as to cross-layer designs [4,5,6,7]. Detailed studies about existing testbeds regarding to hardware architectures and software components are presented in [8,9]. References [10,11,12] survey image processing algorithms in sensor networks.Reliable transport protocols are investigated in [13]. Energy efficient and/or QoS aware routing protocols are compared in [14,15,16]. Respectively, QoS-aware MAC protocols and multi-channel MAC protocols are reviewed in [17,18]. While T. AISkaif etc. provide detailed energy models and numerical studies for MAC protocols in low data rate WMSNs in [19]. References [20,21] analyze cross-layer solutions for multimedia delivery over sensor networks. References [22] provides an insight into the energy problem that concerns all sensor networks, and [23] in specific discusses energy efficiency with regard to target tracking applications. Finally, security issues and privacy problems are summarized in [24,25].

Our work differs from previous surveys mainly in the following three grounds. Firstly, we outlined the four nuclear characteristics of WMSNs: Battery-powered, real-time video data, large volume of sensory data, and directional sensor coverage. For each characteristic, we couple it with the respective requirements and corresponding solutions in the hope that it will aid researchers from different areas to better grasp the challenges in their own field and help to clarify what needs to be done in their particular research directions. Moreover, we are also the first to give formal definitions to local processing, multimedia in-network processing, and camera calibrations. Secondly, to provide network planners and policy makers with a clearer overall understanding of WMSNs, we present the characteristics of WMSNs together with the imposed requirements and the existing solutions in a novel way such that all of these aspects are illustrated as an inter-connected unibody instead of scattered fragments. Last but not the least, we survey state-of-the-art energy-efficient and QoS-aware communication protocols including MAC protocols and routing protocols. To be precise, MAC protocols with prioritization and service differentiation functionalities and disjoint multipath routing protocols are studied. For each topic we not only sum up the pros and cons of each surveyed protocol but also we provide a look into its future trends. To the best of our knowledge, our work is the first to achieve all three at the same time.

The rest of our paper is organized as follows. The characteristics of WMSNs are discussed in Section 2, along with the imposed challenges and respective existing solutions. Section 3 studies MAC protocols designed for WMSNs while Section 4 investigates a series of disjoint multipath routing protocols. Lastly, we conclude our work in Section 5.

## 2. Wireless Multimedia Sensor Networks

As an emerging technique for providing multimedia services, WMSNs possess many unique characteristics when compared with traditional WSNs. However, as shown in Table 1, although these characteristics introduce new capabilities and open doors to new applications, they also raise many new requirements that demand for new solutions. In this section, we discuss these unique characteristics of WMSNs and the relevantly imposed requirements and their solutions.

### 2.1. Battery-Powered

In WMSNs, camera sensors are prevalently powered by irreplaceable and non-rechargeable batteries. As a result, the operation time of a camera sensor is strictly restricted by its energy consumptions from basic system operation, multimedia data computation, and data communication [105]. To ensue a long lifetime of WMSNs, many actions are required. First, designing low-power camera sensor platforms is needed. Second, it is necessary to develop energy efficient computation algorithms and communication protocols to cut down energy drainage caused by the two main culprits, i.e., computation and communication. Lastly, to achieve an optimal energy usage pattern dynamic power management policies can be applied.

#### 2.1.1. Energy-Efficient Computation

In traditional WSNs, a well-known fact is that the level of computational energy consumptions can be neglected. However, it is shown in [5] that the energy used for computation in WMSNs is intrinsically high. For example, in a simple vehicle tracking application, the energy for frame capture and processing can reach up to 12% of the overall energy consumption in the detection of an event [61]. Consequently, WMSNs must try to utilize the more energy-efficient vision processing algorithms, such as energy-efficient image processing [26,27,28,29,30,31] and energy-efficient video compressing [32,33,34]. Moreover, whilst pursuing low level of computational energy usage, a tradeoff between energy consumption and image quality must also be addressed [106,107].

#### 2.1.2. Energy-Efficient Communication

Like traditional WSNs, data communication also accounts for the majority of energy consumption in WMSNs. In fact giving that there is significantly larger quantity of data to transmit, it is of great importance to have communication protocols designed to be highly energy-efficient. This applies to every communication layer in the protocol stack. For example, transport protocols can adjust the event reporting frequency according to the level of reliability achieved [38], routing protocols can use energy prediction technics and load balancing mechanisms to even energy distribution across the network [108,109], and MAC protocols can accommodate dynamic duty cycle to conditionally put some nodes to sleep so as to avoid idle listening [40,41]. Furthermore, a tradeoff between computation and communication power usage can also be derived to minimize overall energy consumption [110].

#### 2.1.3. Dynamic Power Management

Dynamic power management (DPM) refers to the exploitation of idleness of different system parts of a sensor node in order to perform selective shutdown of idle components [35]. By keeping track of energy consumption at different components and taking into account the network dynamics, a DPM framework can derive policies to decide when a component can be shut down or when a sensor node can go to sleep and how long the component or sensor node stays in the hibernation state. For WMSNs, a major challenge in designing DPM frameworks is the necessity to consider QoS guarantees. The work by Fallahi and Hossain [36] is a good example of QoS-aware DPM framework. It uses a Markov decision process to build the DPM framework by taking into consideration the operation states of the camera and transceiver, the network dynamics including queuing status and channel condition, and the video traffic arrival process. And a dynamic programming approach is used to derive the optimum policy of the DPM framework. Nonetheless, one drawback in DPM is the risk of missing events when shutting down the camera. To tackle this problem, Sinha and Chandrakasan [37] proposed an approach—disallow sleep of camera if the task is critical and allow for sleep of camera in non-critical tasks when the sleep threshold is satisfied.

### 2.2. Real-Time Video Data

Most applications of WMSNs require delivery of real-time video data. To put it straight, WMSNs need to send video data to the sink before a deadline and the ratio of packet loss should not exceed a defined threshold. Apparently, stringent QoS demands in terms of delay and reliability will be imposed on WMSNs as a result. Nevertheless, for a resource constrained network with large volume of multimedia data to transmit, providing guarantees on such QoS requirements is rather a challenge. In order to sum up existing solutions, here we outline the fundamental ways of achieving QoS provisions.

#### 2.2.1. Delay

For applications like security surveillance and traffic monitoring, timeliness is key in the delivery of the captured video data. Any miss of deadline means system failure and can cause security breach or traffic incidents. Translate this into network operations, the meanings could be twofold. On one hand, MAC protocols can utilize prioritization and service differentiation schemes to give real-time video data higher priority for earlier access of the wireless medium, or grant it with access to higher quality channels. For example data with higher QoS demands are assigned with the earliest available slots with higher quality channels in [42]. On the other hand, routing protocols can deploy QoS aware mechanisms to choose paths that are able to meet the end-to-end delay requirement. For example choosing the path with the least accumulated delay along the traversed hops [47].

#### 2.2.2. Reliability

Reliability is another important QoS measure for real-time video delivery over the lossy wireless medium. Ideally, all video data needs to be delivered to the sink successfully since data loss could result in missing events of interest. In general, reliability guarantee can be provided either by redundancy or by retransmission [13], as shown in Figure 1.

Depending on how redundant data is introduced, we classify redundancy oriented reliability into bit-level redundancy and packet-level redundancy. In bit-level redundancy, extra bits or segments are added to an existing packet. In case the packet is damaged or some segments are missing, the receiver can use the redundant bits to recover the packet or utilize the redundant segments to reconstruct the packet. Regular bit-level redundancy techniques include forward error correction (FEC) and erasure codes. On the other hand, packet-level redundancy generates redundant and standalone copies of packets. There are two commonly used techniques for packet-level redundancy. The first is multipath routing, where redundant copies of the same packet are sent from the source via multiple disjoint paths to the sink. Naturally this increases the chance that at least one copy of the packet is successfully delivered, therefore improving reliability. The second is by applying random linear network coding (RLNC) [52,53]. RLNC creates *m* coded packets out of every *n* native non-coded packets with (m≥n). As long as *n* out of *m* coded packets are correctly received at the destination, all of the native packets are guaranteed to be successfully received after a decoding process.

Another way for providing reliable data delivery is to retransmit the lost or damaged packets. Communication protocols can either introduce redundancy to packets or deploy mechanisms for retransmission to guarantee reliable communication. However, it is worth to note that the cost of introducing redundancy and retransmission must be considered since sending extra bits or redundant packets requires more resources.

#### 2.2.3. Prioritization and Differentiated Service

WMSNs encompasses both camera sensors and a variety of scalar sensors. As a result, heterogenous types of traffic including video, audio, and scalar data often coexist at the same time in WMSNs. This no doubt adds to the complexity of QoS provision because each type of traffic possesses distinct attributes and has different QoS requirements. Moreover, since WMSNs are no longer employed for a singular application, every application will also have its very own QoS demands. Even for the same application, however, different events may as well require different QoS guarantees. Not to mention there are extreme cases where different packets can have different QoS requirements. For example, packets containing I-frames in a video stream have higher QoS demands than packets with B-frames and P-frames. To tackle these issues, communication protocols needs to consider the usage of prioritization schemes which assign separate priority levels to different traffic types (or different applications, events, and packets) for providing differentiated services [4]. Commonly, there will be a classifier to assign proper priority levels to the packets and a scheduler to determine which packets should be first served according to the assigned priority levels [54]. With the priority levels set for each packet, routing protocols and MAC protocols can adjust their behavior in ways such that packets with higher priority will be sent on a shorter path, or be given earlier access to the medium.

#### 2.2.4. Quality of Experience

Quality of Experience (QoE) is a metric used to evaluate an end-user’s satisfaction on the perceptions for the quality of multimedia services [55,56]. In contrast to the aforementioned QoS provision methods that emphasize on technical terms for packet delivery, QoE stresses on the guarantee of a minimum level of quality of the real-time video data in order to meet the end-user’s demands. As a result, providing QoE guarantees often requires cross-layer cooperations, from multimedia encoding in the application layer, to the QoS provisions in the communication layers. For example, in [57] a joint encoding and routing approach is studied which explores the correlation characteristics of video data in WMSNs to reduce data redundancy while also to improve frame delivery ratio. In [58] a cross-layer framework that jointly considers QoS and QoE is designed by using scalable high efficiency video coding (SHVC) with error concealment and multipath routing.

### 2.3. Large Volume of Multimedia Data

The volume of multimedia sensory data in nature exceeds that of classical scalar sensory data on a scale of magnitudes. For WMSNs with which the scarcity of bandwidth is norm, transmitting large volume of multimedia data is quite an obstacle to overcome. First of all, methods to reduce the amount of data to be transmitted are necessary in order to cut down bandwidth requirement. Then mechanisms to maximize the utilization of the available bandwidth has to be explored. If possible, WMSNs have to adopt technologies that can potentially provide higher available bandwidths.

#### 2.3.1. Reducing Data Redundancy

With limited bandwidth available, having WMSNs to deliver all of the captured multimedia data is certainly not cost-effective. As a result, it is mandatory to introduce ways to reduce the volume of data to be transmitted. For example excessive redundant data can be removed by applying onboard local processing algorithms, or by implementing distributed source coding and multimedia in-network processing amongst neighboring nodes. Meanwhile, possible in-network data storage solutions can also be used to limit the necessity of data transmissions.
Local Processing: Local processing refers to using onboard image analysis techniques to extract useful imagery components for the description of events of interest [59,60,61]. Depending on the intelligence of algorithms, local processing can be categorized into different levels [60], in which the required bandwidth decreases with the increase of algorithm intelligence and level of inference, as shown in Figure 2. One example is the large vehicle detection scenario described in [61]. Instead of streaming raw video data, a camera sensor in its basic setting transmits a whole frame to the sink whenever motion is detected. However when more local processing is allowed, camera sensors are able to perform background subtraction to detect the moving object. Only if the detected object is larger than a threshold then the portion of image containing the detected object will be transmitted. With more intelligence and coupling with multimedia in-network processing, camera sensor nodes can cooperate, and a portion of the image with the detected object will only be sent when two camera sensors have determined the object is of interest following their own individual criteria. Furthermore, as demonstrated by Zhai et al. [62], camera sensors can collaborate and produce a textual only description for the events. Although local processing is a promising way to reduce the amount of data to be transmitted, it is worth to note that however, accompanying local processing are that more hardware resources will also be needed.Multimedia In-Network Processing: Multimedia in-network processing is defined as intermediate network computing amongst local nodes to promote network scalability through energy savings [111]. Under the current literature, there exists mainly two kinds of multimedia in-network processing techniques as given in Figure 3. The first is multimedia data fusion where typically a cluster head gathers data from its cluster members (normally after a certain level of local processing has been executed) and combines them to create a summarized report on the event of interest so as to reduce data redundancy and also to improve level of inference. Due to its sensor heterogeneity, multimedia data fusion in WMSNs usually is multimodal, i.e., involving fusion of different types of data such as images, videos, and non-imagery data [63,64]. Online multi-view video summarization [65] is another in-network processing technique. For a group of camera sensors with overlapped field of views (FoVs), an online local processing algorithm is used to select the important frames at each sensor and only features of the selected frames together with their foreground mask sizes are broadcast. By comparing the received features and foreground mask sizes with its own, a sensor then decides whether its selected frames will be delivered to the sink or not. From the received frames, the sink does the analysis to obtain the event information.Distributed Source Coding: In typical multimedia source coding problems, all of the source information is available for compression at a centralized place. However, in WMSNs the correlated source information usually resides in multiple camera sensors [66]. Due to resource limitations, transferring all source information to a centralized location is certainly not feasible. One solution is to use distributed source coding (DSC), which allows each camera sensor to independently encode their own piece of source information while leaving the complex joint decoding work to the sink [67,68]. Compared to traditional downlink multimedia source coding technics, such as the JPEG 2000 and MPEG.x, the main advantages of DSC is the ability to shift much computation burdens from the encoder side to the decoder [34]. For WMSNs, this can significantly reduce energy consumption and also lower hardware costs [27,30].In-network Data Storage and Query Processing: Traditionally, WSNs would transmit all sensing results to the sink for further processing and future inquiry. However, it is shown by Y. Diao et al. [69] and H. Li et al. [70] that with recent technology advances, it is now practical to equip camera sensors with more powerful processors and significantly larger flash memories for local processing and data storage. On one hand, with data being processed and stored on site, only the end result of data analysis needs to be sent to the sink. On another, in case of historical data inquiries, the query request (e.g., the number of traffic incidents happened at an intersection during the past year) can be pushed into the network and only the query results needs to be transmitted. In both scenarios, data transmission can be reduced and energy cost is also lowered. Nevertheless, local data storage schemes do impose a couple of challenges. First, when local storage space starts to fill up, proper data aging processes are required to clear space for new data without sacrificing fidelity on critical historical data [69,71]. Second, sensor nodes in this scenario essentially form a distributed probabilistic database, thus respective database management techniques, have to be considered for efficient data querying [72,73].

#### 2.3.2. Higher Bandwidth Requirements

Bandwidth has always been a scarce resource, and definitely more so in WMSNs. To illustrate, the nominal transmission rate of the IEEE 802.15.4 compliant camera sensor motes, such as MeshEye [60], iMouse [112], and Vision Mesh [113], is only 250 Kbit/s. But the required bandwidth for general multimedia applications could be at least one order of magnitude higher [4]. What is more, the need to relay multimedia data to the sink and the coexistence of scalar data traffic can easily raise the bar further on bandwidth requirements [54,114]. In order to overcome the challenge, it is necessary to develop communication protocols that can utilize bandwidth more efficiently. To begin with, through the usage of multichannel MAC protocols, a group of nodes can communicate in parallel over different channels [74,75,76]. Instead of adopting a single routing path for data delivery, using multiple independent paths simultaneously to split the traffic will allow to achieve higher aggregated bandwidth [54]. In the meantime radio technologies with potentially higher bandwidth, such as the ultra-wideband (UWB), can also be used in WMSNs [77,78].

### 2.4. Directional Sensor Coverage

Unlike the omnidirectional scalar sensor siblings, camera sensors in WMSNs are directional. As shown in Figure 4, the coverage regions, namely FoVs, of camera sensors are determined by two factors: Intrinsic camera parameters such as lens focal length, pixel location of the optical center, and skew factor [79,115]; and extrinsic camera parameters like location and orientation [90]. The directional FoVs of camera sensors brings many new challenges upon WMSNs. First, both intrinsic and extrinsic camera parameters need to be calibrated accurately. Because correct positioning of cameras and accurate interpretation of FoVs are key to the the success of all monitoring tasks. Second, even though WMSNs are usually deployed with a redundant number of camera sensors, means of maximizing the coverage region with only a minimum number of active camera sensors are necessary in order to conserve energy [116]. Last but not the least, as many applications will often require collaboration among multiple camera sensors, careful designing of collaboration schemes is required.

#### 2.4.1. Accurate Camera Calibration

Camera calibration is necessary for mapping imagery metrics and coordinates into real world metrics and coordinates. Depending on the parameters being calibrated, camera calibration can be classified into intrinsic camera calibration and extrinsic camera calibration as summarized in Figure 5.
Intrinsic camera calibration (ICC): ICC refers to the computation of internal camera parameters for transforming imagery metrics to real world metrics, for example pixel distance to real world physical distance [79,80]. Depending on the assumptions made upon the scene structures used for calibration, ICC can be carried out either with assistance of reference targets or target-free under self-calibration [81]. Oftentimes reference targets are used when it is important to obtain high accuracy of calibration [82]. Common targets placed in the scene can be 1D objects [83], 2D objects [84], and 3D objects [81], with calibration accuracy and complexity both increase as the targets get more sophisticated. Self-calibration, on the other hand, requires no reference targets but rather relies on the static scene structure and camera motion [85]. With no assumptions made on the scene structure, however, self-calibration needs to estimate a large number of parameters and quite often the correlation between these parameters lowers accuracy of calibration [86].Extrinsic camera calibration (ECC): With ECC, we can calculate the external parameters of a camera sensor. Such information is used for converting imagery coordinates to real world coordinates [80,87], which is crucial in order to make the imagery data geographically meaningful. The decisions on ECC can be told apart from three criteria. First, depending on where the calibration actually takes place ECC can be carried out either in a centralized or a distributed fashion [79]. When centralized ECC is executed, imagery tracking data captured by all camera sensors are sent to the sink, with which the sink calculates the extrinsic parameters for all [88]. In distributed ECC however, each camera sensor determines its own extrinsic parameters [79,80,89]. Second, judging by whether overlapped FoVs exists or not different ECC approaches can be deployed [90]. When there exists overlapped FoVs, pairwise calibration is usually used first to find correspondences in the overlapped FoVs. An optimization procedure is then adopted to obtain the best matching correspondence [91,92]. On the other hand, when no overlapped FoVs present, prior knowledge on the moving target (e.g., velocity) and trajectory prediction are needed for calibration [88,93]. Third, depending on the degree of constraints placed on the calibration environment, ECC can either be supervised or fully automated. Most ECC techniques under current literature fall in the supervised category. They demand a priori of the target, such as location of the target [79,93], height of the target [92], or the target’s motion dynamics [88,93,117]. On the contrary, in fully automated ECC, no assumptions on the target or the environment are necessary [87,94].

#### 2.4.2. Coverage Optimization

Coverage optimization in WMSNs is different from that in traditional WSNs due to the directional coverage of camera sensors [95,96]. The objective of it also strongly depends on the application scenarios. In target tracking applications, coverage optimization could be aimed at providing full angle view of the target with a minimum number of camera sensors [97]. Or it could be formulated to cover all targets of interest in the field [98,99]. In intruder detection applications, the camera sensors are optimized to form a coverage barrier in order to detect any trespassing intruders [100,101]. Whilst in surveillance applications, coverage optimization is aimed at providing occlusion-free and full area coverage with minimum overlapped FoVs [95]. Regardless of the objectives, however, one common goal is to provide prolonged and continuous monitoring. Hence it is necessary to maximize the lifetime during which the coverage requirement can be satisfied through careful duty cycling of camera sensors [102].

#### 2.4.3. Camera Sensor Collaboration

Because of the individually limited FoVs, applications such as continuous tracking would require collaboration among multiple camera sensors. Figure 6 shows how K. Obraczka et al. [103] classify camera sensor collaboration schemes. Based on whether overlapped FoVs exists or not, the authors distinguish camera senor collaboration schemes into spatial-based collaboration where overlapped FoVs exist, and predictive collaboration in case of the opposite. In spatial-based collaboration, a graph of overlapped FoVs needs to be constructed and camera pairs with overlapped FoVs ought to be jointly calibrated and synchronized. Instead for predictive collaboration, neighborhood camera sensor distributions and target motion information are required to predict into which camera sensor’s FoV the target is going to move. Regardless of the collaboration schemes, one common task to establish efficient camera sensor collaboration is to carefully design task handover procedures as described in [104].

### 2.5. Conclusion

We have presented in detail the characteristics and requirements of WMSNs. With each requirement, we also discuss its existing solutions. To conclude our introduction to WMSNs, we show in Figure 7 as to how its characteristics and requirements are related. We believe such an illustration will act as bridge among researchers in different fields and provide them with the big picture needed for such an interdisciplinary subject. As shown in the figure, the core of WMSNs are the battery powered camera sensors. These camera sensors possess unique directional sensor coverage and generate large volume of multimedia sensory data. First, the existence of directional sensor coverage requires camera sensors to be accurately calibrated so as to make the captured imagery data geographically meaningful. At the same time, in order to successfully accomplish the monitoring tasks, coverage optimization and camera sensor collaboration are also needed. Second, as it is impractical to send all multimedia data to the sink, redundancy within the data should be minimized. The common approaches to achieve this is to use local processing, distributed source coding, multimedia in-network processing, and in-network data storage and query processing techniques. However, even after redundancy is reduced, the amount of data to be transmitted is still at large compared with traditional WSNs. Therefore much higher bandwidth is required. Moreover, within the multimedia data, there includes a big portion of real-time video data that calls for stringent QoS guarantees. On the other hand, due to the battery powered nature, the lifetime of camera sensors directly limits the lifetime of WMSNs. As a result, energy efficiency is required for all computation and communication done by the camera sensors for they accounts for the majority of energy consumptions. For example, computation intensive tasks such as local processing and distributed source coding needs to be done energy efficiently; communication protocols designed to achieve higher bandwidth and/or to provide QoS guarantees should be energy-aware; and protocols such as camera sensor collaboration, multimedia in-network processing and etc., that require both computation and communication, need to have energy efficiency designed in mind as well. Last but not the least, dynamic power management can also be adopted to manage the internal operations of a camera sensor in order to further conserve energy.

## 3. QoS Aware MAC Protocols for WMSNs

Designing energy efficient MAC protocols to coordinate the transmission of large amount of sensory data and to meet the stringent QoS requirements for WMSNs is a very challenging task. First of all, although duty cycling of radio is a common MAC layer practice to save energy, extra caution should be taken when applying the same technique for MAC protocols in WMSNs due to the dynamic and bursty nature of multimedia traffic. Secondly, in order to guarantee the high data rate needed to transmit the large volume of data, MAC protocols should be designed to be able to minimize collisions, especially in times of transmission of real-time video data. Thirdly, due to the coexistence of heterogeneous types of traffic, a prioritization and service differentiation mechanism is required to allow earlier access to the wireless medium for higher priority traffics.

In this section we investigate energy efficient MAC protocols which can provide QoS guarantees by means of prioritization and service differentiation. We summarize the surveyed MAC protocols in Table 2.

### 3.1. EQ-MAC

Yahya and Ben-Othman proposed a hybrid energy efficient and delay-aware MAC protocol named EQ-MAC [39]. In EQ-MAC, sensors are grouped into clusters where the cluster head takes care of scheduling TDMA based slots for data transmissions. Communication in EQ-MAC is organized into frames. Each frame begins with a synchronization message broadcast by the cluster head. After synchronization, cluster members enter a CSMA/CA based medium access phase to transmit their control messages to the cluster head and those with data to transmit will also send requests for TDMA slots. Upon receiving all requests, the cluster head generates a schedule of TDMA slots with consideration of traffic priorities and the schedule is broadcast to all cluster members for the subsequent data transmission phase. Cluster members transmit data during their giving slot in a contention-free manner while nodes with no data to send go to sleep to save energy. Delay guarantee is achieved by classifying packets according to their priority levels assigned in the application layer, and real-time packets are stored in an instant queue in which packets will be served immediately.

Pros and Cons. By allowing nodes without data to transmit to go to sleep, EQ-MAC achieves energy efficiency and at the same time improves channel utilization under light traffic conditions. Meanwhile, by giving definite priority to high QoS required instant traffic, EQ-MAC is good for delivering multimedia data that requires minimum delay. However, this strategy can lead to starvation of low priority traffic such as best effort traffic. Also the necessity of global synchronization makes the protocol not scale well.

### 3.2. N. Saxena et al. and Diff-MAC

Saxena et al. [40] and Diff-MAC [41] are two similar CSMA/CA based QoS aware MAC protocols with adaptive contention window (CW) and dynamic duty cycling. In order to give real-time traffic precedence for channel access, both protocols set the CWmin and CWmax of real-time traffic to be smaller than lower priority traffics. In the meantime, in times of collision CW of real-time traffic increases slower as compared to other traffic types and decreases faster when network condition recovers. The differences are that Saxena et al. [40] uses the “stop-for-a-round” method for CW adaptation in pursuance of fairness among sensors. To be precise, a sensor will seize to adjust its CW size and wait for neighboring sensors to adjust their CWs, if it sees the collision probability not changing as expected after it has last adjusted its CW size. On the other hand, Diff-MAC allows sensors to continuously adjust their CWs for fast convergence to the target CW size.

Besides CW adaptation, Diff-MAC implements a hybrid weighted fair queuing (WFQ) method to further reduce channel access delay for real-time traffic. Unlike the FIFO scheduler used by Saxena et al. [40], WFQ method applies different weights on separate queues with the queue for real-time traffic taking the bigger share of total weight. The scheduler then chooses the packet to be served next in virtue of weights on the queues. Moreover, other than prioritization among different traffic types, Diff-MAC also prioritizes packets within the same queue by sorting them based on their traversed hop counts so that packets which the network has invested more resource in will get served first since dropping such packets will yield greater waste. Diff-MAC also adopts the message passing technique to fragment video frames into smaller packets and send them in a burst in order to reduce retransmission cost.

Another technique the two protocols have in common is the dynamic duty cycling mechanism. By monitoring the statistics of processed packets (both sent and received) a sensor determines the dominant type of traffic and accordingly configures its active period. When real-time traffic is dominant, active period is extended to ensure minimum idle waiting time at the expense of higher energy usage. Whereas if non real-time traffic or best effort traffic is dominant, active period is reduced to avoid idle listening and save energy. This allows the network to balance delay and energy consumption while at the same time to adapt to the dynamic traffic patterns of WMSNs. Although the lack of synchronization can lead to suffering of early sleeping problem.

Pros and Cons. In terms of QoS provision, energy efficiency and fairness, both Saxena et al. [40] and Diff-MAC [41] are good candidates. However, it is quite costly since the two MAC protocols require sensors to constantly monitor many network states. The lack of sleep-listen synchronization leaves the network prone to idle listing and early sleeping problem considering the dynamic and bursty traffic in WMSNs. As for Diff-MAC, the continuous intra-queue sorting also becomes infeasible when the traffic load is high.

### 3.3. MQ-MAC

MQ-MAC [42] is a slotted CSM/CA based MAC protocol for cognitive radio sensor networks. In MQ-MAC, nodes form clusters and the key procedures including channel sensing, channel assignment, and guaranteed time slot (GTS) allocations are all coordinated by the cluster heads.

Like IEEE 802.15.4, the superframe of MQ-MAC also composes of an active period and a sleep period, and the active period is further divided into three phases. The cooperative sensing and channel selection phase, where cluster members assign weights to the polling channels $S_{K}$ according to the detected channel status (idle, busy, collision) and results of channel sensing are reported to the cluster head. Next the cluster head uses a weighted moving average method to calculate the average weights $WA_{c}$ for each channel and accordingly organize the channels into a decreasingly ordered set $C_{b}$. Together with the channel sensing results, data transmission requests are also sent to the cluster head using CSMA/CA in a manner that nodes with traffic of more stringent QoS requirements are given smaller back-off counters for earlier medium access.

After obtaining the channel sensing results and data transmission requests, the cluster head then performs slot allocation and channel assignment. In MQ-MAC, slot allocation is only for QoS required traffic and it follows three basic rules. First, for each transmission request with a packet arrival rate $n_{p}$, $n_{p}$ GTS slots are allocated. Second, transmission requests are classified by their traffic types, and within each traffic type the requests are ascending ordered in accordance with the lifetime of packets. Third, earlier GTS slots are given to requests with higher QoS demands. Before channel assignment, candidate channels $C_{b}$ are separated into three groups such that:(1)$$B=\{c\in C_{b}|WA_{c}\geq \left(\mu +\sigma \right)\}$$
(2)$$M=\{c\in C_{b}|\left(\mu -\sigma \right)\leq WA_{c}\leq \left(\mu +\sigma \right)\}$$
(3)$$D=C_{b}-B-M$$
where μ and σ are the mean and standard deviation of $WA_{c}$, *B* is the best channel set, *M* is the moderate channel set and *D* is the group of channels that will be discarded due to poor quality. First, channels in *B* are each assigned with multiple GTS slots, then a single GTS slot is assigned to every channel in *M*. This channel assignment process carries on iteratively until all slots are designated with a data channel. Moreover, for each slot the next best channel following the assigned data channel is also chosen as a backup. The same channel assignment method is also used for best effort traffic.

Following the above slot allocation and channel assignment technique, QoS required traffic can be sent to the cluster head without collision in the data transmission phase. On the contrary, best effort traffic are transmitted using CSMA/CA and packets with shorter remaining lifetime are giving higher priority of medium access. After data transmission nodes enter a sleeping state until the next superframe begins.

Pros and Cons. MQ-MAC provides QoS support by giving different types of traffic distinct priorities in terms of slot allocation and channel assignment. However the protocol is not well fit for the dynamic and bursty traffic in WMSNs due to the fixed superframe size and the fact that a node may be required to switch among different channels for a single transmission session. Moreover, using traditional CSMA/CA for the communication among cluster heads could jeopardize the whole QoS provision scheme. Also the protocol overhead is high due to frequent exchange of control messages, channel sensing and channel switching costs.

### 3.4. IH-MAC

M. Arifuzzaman et al. presented an intelligent hybrid MAC protocol IH-MAC [43]. Unlike EQ-MAC [39] that has CSMA/CA and TDMA operating individually under different network conditions, IH-MAC fuses the key concepts of CSMA/CA and TDMA together to create a new MAC mechanism. First of all, instead of global synchronization, IH-MAC adopts local neighborhood synchronization from S-MAC [118]. Another, although IH-MAC operates in slotted mode like TDMA, not all slots are contention-free and the contention-free slots are scheduled in a decentralized fashion. In fact, nodes claims ownership of slots through clock arithmetic:(4)s≡i(modn)
where *s* is the slot number, *i* is the node ID, and *n* is the number of neighbors within a two-hop range. However, ownership of a slot does not guarantee absolute transmission opportunity. By using CSMA/CA with non-overlapped contention windows, the protocol gives unquestionable priority to the node possessing data of high QoS demand. If there is no such critical data then owners that have their IDs mapped to the same slot can contend for that slot. Only when previous cases do not apply will non-owner nodes have the chance to contend for the slot. A transmission power adjustment technique is also used in the contention-based period to conserve energy. On the other hand, contention-free slots are only claimed when traffic load is high. If the buffer size of a particular node exceeds a certain threshold, the node makes some of its owned slots into rendezvous slots using another clock arithmetic:(5)$$s_{r}\equiv i\;\left(\mathrm{mod}\;m\right)$$
where $s_{r}$ is the reserved slot number, *m* is multiple of *n*. Information about the rendezvous slot is broadcast in the neighborhood and during the rendezvous slot neighbors of both sender and receiver go to sleep. As a result, no contention is required and only DATA and ACK are exchanged in a rendezvous slot just like a nominal TDMA time slot.

Pros and Cons. By fusing CSMA/CA and TDMA together, IH-MAC avoids the lack of scalability from traditional TDMA while also reducing collision and improving channel utilization and access delay that are main drawbacks of legit CSMA/CA. Nevertheless, the need for managing a two-hop neighbor list adds extra cost to the protocol, and the loosely synchronized nodes can end up with early sleep problem.

### 3.5. AMPH

AMPH [44] by M. Souil is similar to IH-MAC [43]. The major difference is that IH-MAC is mainly CSMA/CA based while AMPH is mainly TDMA based. Communication time in AMPH is divided into slots and within a two-hop range each node is assigned a distinct slot. Nevertheless, a node is allowed to transmit in any slot even if it is not the owner. This is done by allowing non-owner nodes to contend for slots using the backoff mechanism similar to CSMA/CA. By separating real-time traffic from best-effort traffic and judging whether a node is an owner or non-owner of a slot, the contending nodes are divided into the following four groups, as listed in an order of decreasing priority: Owner node with real-time traffic, non-owner node with real-time traffic, owner node with best-effort traffic, and non-owner node with best-effort traffic. Higher priority nodes are granted with earlier medium access with the non-overlapping contention window technique. To be precise, each node chooses a random backoff timer from the respective window at the beginning of a slot. When the backoff timer expires, nodes use clear channel assessment (CCA) to check the channel status. The node that wins channel contention sends data using the message passing feature, while nodes fail channel contention wait until the next slot. To be fair, AMPH allows best-effort traffic to have higher priority over real-time traffic in a few slots during each cycle. At the end of each slot, nodes enter a waiting state during which the radio can be switched off to conserve energy.

Pros and Cons. AMPH achieves high channel utilization and good adaptability to varying traffic loads by allowing nodes to use any slot for communication. The dynamic priority scheme also provides QoS support and good fairness among heterogeneous traffic types. However, the energy efficiency design in AMPH is inferior since it requires a node to keep listening to the medium until the waiting state even if it does not involve in any data communications. The simple differentiation between real-time and best-effort traffic may be inefficient for WMSNs with more types of coexisting traffic.

### 3.6. PA-MAC

PA-MAC [45] is an IEEE 802.15.4 based multi-channel MAC protocol. The superframe structure of PA-MAC is similar to IEEE 802.15.4. But a dedicated channel is used for the beacon frame for transmission of control information such as GTS slot request, traffic priority class, and data channel assignment. PA-MAC classifies traffic into these four categories in a decreasing order of priority: Medical traffic including emergency traffic, on-demand traffic, normal traffic, and non-medical traffic such as audio and video. Accordingly, the contention access period (CAP) is sequentially divided into four sub-phases. In favor of medical data, nodes with higher priority traffic is granted with access to the slots in the sub-phases for low priority traffic. The length of each sub-phase is controlled by the proportion of nodes with respective category of traffic. As for nodes with on-demand traffic and non-medical traffic, their CAP sub-phases are only used to send GTS slot requests to the coordinator while the actual data transmission takes place in the contention free period (CFP). After data transmission, nodes enter a sleep state until the next superframe begins.

Pros and Cons. PA-MAC is proposed for wireless body area networks (WBANs) that can be considered as a sub-category of WMSNs in which multimedia data has lower priority instead. PA-MAC reduces the collision ratio of traditional IEEE 802.15.4 by using service differentiation, and by transmitting continuous multimedia data in the CFP period. However, the collision ratio is still high compared to unslotted CSMA/CA, especially when node density increases, and the protocol is not suitable for regular WMSNs.

### 3.7. Conclusion and Future Trends

To design MAC protocols for WMSNs, one must take into account the existence of heterogeneous traffic and the nature of bursty and voluminous multimedia data traffic. We found that in the current literature there exists mainly three types of MAC protocols depending on the underlying medium access mechanisms. The CSMA/CA based protocols [40,41] boost good scalability and adaptability to varying traffic conditions as it requires no synchronization. However, CSMA/CA suffers from lacking of QoS support and energy inefficiency due to its high collision rate and high control overhead especially when under heavy traffic. This can be alleviated by using traffic type prioritization and service differentiation techniques such as adaptive contention window and dynamic duty cycling. The hybrid protocols of CSMA/CA and TDMA [39,43,44] in essential provide better QoS support and energy efficiency while at the same time maintain a low control overhead. With dynamic slot allocation techniques, the hybrid protocols are also more adaptive to dynamic traffic conditions compared to traditional TDMA. Nevertheless, channel under utilization still exists when traffic is light and scalability of the hybrid protocols also needs to be further improved. On the other hand, the slotted CSMA/CA (IEEE 802.15.4) based protocols [42,45] often employ a mutli-channel design, with a dedicated channel for control messages and a set of data channels for data communications. However, although the multi-channel design improves channel efficiency, it also brings on channel switching costs. And a complete GTS slot based protocol [42] lacks scalability and adaptability to the dynamic traffics, while an IEEE 802.15.4 based protocol suffers from high collision rate. Despite the respective pros and cons of each category of MAC protocols, it is to be noted that, however, there exists the common drawback of these MAC protocols being not considering the nature of multimedia data characteristics.

To accommodate the bursty, heterogeneous, and voluminous varying traffic patterns, and to guarantee QoS provision, we believe the best MAC protocol for WMSNs should be a hybrid of CSMA/CA and TDMA with traffic prioritization, dynamic slot allocation, and adaptive contention window mechanisms without imposing high energy cost. Furthermore, to reduce channel access delay and to better accommodate multimedia traffic, the message passing technique should also be considered.

## 4. QoS Aware Multipath Routing Protocols for WMSNs

Routing strategy is perhaps the most studied subject in WSNs concerning communication protocols. However, routing protocols for traditional WSNs can not be directly applied in WMSNs. For one thing, in traditional WSNs routing protocols are designed to deliver scalar data over a single shortest path. Nevertheless, sending a large amount of multimedia data over the shortest path will likely cause severe network congestion and end up with early node death. Therefore using multipath routing techniques to distribute multimedia traffic over multiple concurrent paths seem to be the natural approach. On anther, in traditional WSNs the main focus of routing protocols is energy efficiency while little to no QoS provision is offered. But in WMSNs, QoS is a major concern as a matter of fact that multimedia data needs to be sent in real-time and reliably.

In this section we present a brief survey of multipath routing protocols with QoS guarantees. A comparison of the surveyed protocols is given in Table 3. Although a few related surveys on multipath routing protocols have recently been published [119,120,121], however our survey studies a largely complete different set of protocols. At the same time we practice the methodology that behind every multipath routing protocol there is a classic single path routing.

### 4.1. DGR

M. Chen et al. proposed a directional geographic multipath routing protocol DGR [46]. The main idea of DGR can be explained with Figure 8. When a node *j* receives the broadcast probe from the upstream source node *i*, node *j* converts its absolute coordinates $\left(x_{j}^{0},y_{j}^{0}\right)$ in the global coordinate system to the virtual coordinates $\left(x_{j}^{v},y_{j}^{v}\right)$ in the virtual coordinate system. The virtual coordinate system is chosen with *i* located at the origin and the x-axis connecting *i* to the sink *s*. To control direction of the path, the source node *i* also specifies a deviation angle α. Accordingly, node *j* calculates its mapping coordinates $\left(x_{j^{\prime }}^{v},y_{j^{\prime }}^{v}\right)$ by shifting its virtual coordinates in a clockwise direction if α is positive and counter clockwise direction if negative. Node *j* falls in the forwarding candidate set only if the mapping coordinates lands in the shaded region, from which *R* is the transmission range, $L_{t}$ is the optimal mapping location, and *D* is the threshold. To compete for the role of nexthop, candidates set up timers such that the candidate whose mapping coordinates with smaller distance Δ*D* to $L_{t}$ will get smaller timer. The node, say node *j*, whose timer expires first sends a reply message REP to the upstream node *i*. Upon receiving the REP, *i* responds with a confirm message SEL. Candidates that overhear either the REP or the SEL message will cancel their timers. To ensure path disjointness, the winner candidate *j* will not participate in the establishment of any other paths for the same source node. Node *j* repeats the above procedure by sending its own probe message with an adjusted deviation angle so that the path will gradually converge toward the sink. To establish multiple paths, the source node initiates a series of probe messages, each with a different initial deviation angle. DGR also proposes a video delivery scheme. For each video frame, the source node first broadcasts the entire frame to its one-hop neighbors. Only neighbors that are on the selected paths will then transmit along the respective path a set of video packets specified by the video source.

Pros and Cons. DGR achieves fast and reliable video delivery by using multipath routing and FEC. Owing to its stateless geographic routing, the protocol also provides good scalability. However in times of node failure, the route discovery procedure and the path recovery time is too long. And the lack of consideration for inter-path interference can greatly increase the need of retransmission which in return will result in increased energy consumption. Finally, a more practical scenario needs to be considered instead of allowing only one active video source at any time.

### 4.2. AntSensNet

AntSensNet is a multi-QoS aware ant colony optimization based routing protocol [47]. Operation of AntSensNet is divided into three phases: Cluster formation, route discovery, and data dissemination and route maintenance. During cluster formation a communication backbone with only camera sensors is constructed. The cluster formation begins with the sink releasing a sequence of cluster ants (CANTs). Initial recipients of the CANTs (i.e., cluster heads selected within immediate communication range of the sink) are chosen probabilistically such that camera sensors with more available resources are more likely to be selected, given that the selected camera sensors are located a minimum distance $R_{c}$ away, where $R_{c}$ is the cluster radius. A newly elected cluster head stores the CANT and advertises to its neighbors about its new identity so that later on neighboring nodes that are not chosen as cluster heads can decide to join a cluster. When a cluster head receives a CANT, it is responsible to reduce the time-to-live (TTL) of the CANT, and to choose from its neighboring camera sensors a new cluster head to be the recipient following the aforementioned probabilistic rule. CANTs whose TTL reach zero are destroyed.

After cluster formation, cluster heads begin the route discovery process. Each cluster head is required to manage a routing pheromone table for its neighbors with respect to each traffic class regarding to the four parameters that the protocol focuses on: Energy, delay, packet loss, and memory. In order to find a path to the sink for a specific traffic class, a source cluster head broadcasts a forward ant (FANT). Along the way, the FANTs collect traversed nodes’ IDs together with the accumulative queuing delay, packet loss ratio, available memory, and minimum residual energy of the nodes passed by. When a FANT is received, an intermediate cluster head first updates the information carried by the ant, then randomly chooses the nexthop such that a neighboring node which provides better QoS support and more resources will be selected with higher probability. When the FANTs reach the sink, a route that meets the QoS requirements is selected and a respective backward ant (BANT) is sent on the reserve path. Nodes that receive the BANT updates their pheromone table by increasing the pheromone value on the incoming link of the BANT while decreasing that of the other links. On the other hand, in order to establish multiple paths for video delivery, a special video forward ant (VFANT) is broadcast. The same route discovery process is executed except that intermediate nodes do not discard duplicate VFANTs and multiple video backward ants (VBANTs) are sent by the sink as opposite to the previous single path routing case. It is up to the source cluster head to choose the set of disjoint paths from all of the routes carried by the VBANTs.

When routes are ready, a source cluster head can begin to disseminate data using the maximum probability path. To balance traffic loads under single path routing, the source cluster head periodically sends FANTs to update routes during data dissemination. In times of congestion or link breakage, a maintenance ant (MANT) is generated to inform neighborhood cluster heads to update their pheromone tables and to find alternative routes, regardless to the single path or multipath scenario.

Pros and Cons. AntSensNet provides differentiated end-to-end multi-QoS guarantees for different traffic classes by finding distinct routes for each type of traffic. To better support video transmission, the protocol also extends its route discovery mechanism for multipath delivery. In the meantime, the explicit consideration of residual energy levels in the routing metric and rotation of the role of cluster heads within each cluster also facilitates good energy efficiency. However, AntSensNet requires cluster heads to maintain a routing pheromone table and periodically exchange hello messages which can hinder the protocol’s scalability. And its multipath routing mechanism is only used for video data but not other types of traffic.

### 4.3. Z-MHTR & Z. Bidai et al.

A Zigbee cluster-tree based multipath routing protocol Z-MHTR [48] with interference awareness [49] is studied by Z. Bidai et al. Apart from the conventional Zigbee parent-child tree routing (TR) path (path 1 as shown in Figure 9), Z-MHTR allows source nodes to explore other node disjoint paths using non-parent neighbors.The protocol requires the source node to maintain a record of branches that have been used for TR. A source node *S* begins to build node disjoint paths following three basic scenarios. Suppose the first intermediate node selected by *S* is node $N_{i}$. If the branch that $N_{i}$ is on hasn’t been used for any TR path for *S*, then a node disjoint path can be established from $N_{i}$ to the sink (root node) using TR (path 2 as depicted in Figure 9). If the branch $N_{i}$ located on is already used for a TR path for *S*, then the selection of nexthop depends on the depth $d_{c}$ of the first common node between the TR path starting from $N_{i}$ and the TR path starting from the node that has firstly used $N_{i}$’s branch for TR. If $d_{c}<d_{i}-1$, then the parent node of $N_{i}$ will be selected as nexthop (path 3 in Figure 9). While in case $d_{c}=d_{i}-1$, then the choice of nexthop depends on $N_{i}$’s neighbor statuses. If $N_{i}$ has a neighbor, say node $N_{x}$ whose branch has not been used for TR, $N_{x}$ will be used as nexthop (path 4 in Figure 9). If no such node exists, i.e., all $N_{i}$’s neighbors have had their respective branches used for TR, then $N_{i}$ selects the neighbor node (if it exists) that is not included in any TR path (path 5 in Figure 9). The same rules are executed at following intermediate nodes until the path reaches the sink. And the number of disjoint paths is equivalent to the number of branches in the topology.

To reduce interference among disjoint paths, the authors proposed an interference level metric in their follow up work [49]. Nodes are required to record their list of interfering neighbors, excluding those on the same path, by overhearing the route discovery messages and data packets which are not intended for them. As the route reply message or data acknowledgement travels along the reserve path, the total number of nodes interfering with those that form the path is accumulated. Thus, at the source node, the interference level of a path is computed as the ratio of total interfering nodes to the number of intermediate nodes. And the set of disjoint paths that minimizes the inter-path interference will be selected.

Pros and Cons. Based on the Zigbee cluster tree routing address assignment mechanism, Z-MHTR achieves multipath routing by adding a neighbor table and by maintaining a record of whether or not TR has been used on a branch. In their following work, an interference-aware mechanism is also proposed to mitigate the route coupling problem between multiple paths. Such a stateless and interference-aware design makes the protocol rather energy efficient. However, the protocol is only fit for the Zigbee cluster tree topology, and the number of paths that could be established is inherently limited by the number of branches. Moreover, it does not consider key QoS requirements such as delay and reliability.

### 4.4. GEAM

Instead of using deviation angle like DGR [46] to control directions of multiple paths, the geographic energy-aware non-interfering multipatrh routing protocol GEAM [50] divides the topology into separate districts for individual paths. Precisely, as shown in Figure 10, the topology in GEAM is divided into three areas by virtual coordinates similar to DGR. The source area $A_{src}$ and the sink area $A_{sink}$ are confined by vertical lines located at *R* distance away from the source and sink respectively (*R* is the transmission range). Rows of non-interfering area $A_{ni}$ are in between the two vertical lines with width *R*. Therefore, a district $D_{i}$ consists of sections from all three areas. When sending packets, the source piggybacks the boundary information of the chosen district on every packet. In doing so, intermediate nodes can use the greedy perimeter stateless routing (GPSR) [122] to route the packet through its corresponding district.

To balance loads and energy distribution, GEAM organizes data transmission into runs of fixed lengthes. To guarantee non-interfering multipath routing, each run is further divided into three rounds, in which a district $D_{i}$ belongs to round *j* if $D_{i}\%3=j$. Loads are equally distributed to all districts during the first run. At the end of each run, the sink gathers statistics such as the minimum residual energy level $E_{i}$ among nodes in each district. Such statistics are feedback to the source. Based on the $E_{i}$’s, the source adjusts the utilization rate of each district, consequently the duration of each round in the next run in a way that districts with higher $E_{i}$ will have more loads assigned. In the occurrence of holes, the same perimeter routing in GPSR is used. When the route is reconstructed, the districts are also realigned to ensure non-interfering paths.

Pros and Cons. By dividing network topology into different districts and by organizing data transmission into rounds with mutually distant active districts, GEAM realizes non-interfering routing and at the same time achieves balanced energy consumption and traffic loads. The protocol also scales well with the underlying GPSR routing method. However, the overhead is increased since each packet is piggybacked with border information of the designated district and is required to collect network statistics on-the-fly. The main drawback of GEAM is it does not take into consideration the QoS metrics such as delay and reliability which are important to multimedia delivery in WMSNs.

### 4.5. LCMR

Using the ad-hoc on-demand distance vector routing (AODV) [123], A. Bhattacharya and K. Sinha developed least common multiple based routing (LCMR) [51]. Instead of selecting the shortest path by the number of hops, LCMR uses the routing time (end-to-end delay) to choose multiple paths. In the route discovery process, the source node will accept a route reply message RREP only if it arrives before a given deadline. For each RREP message, the source node identifies the routing time spent by the respective route request message RREQ to reach the destination. Among the accepted *n* paths with routing time $\left\{T_{1},T_{2},\cdots ,T_{n}\right\}$, the least common multiple *L* of $\left\{T_{1},T_{2},\cdots ,T_{n}\right\}$ is calculated. The number of packets sent along a path *i* is then decided such that out of $k=\sum_{i=1}^{n}L/T_{i}$ packets, $L/T_{i}$ packets will be sent along path *i*. Therefore, the total time used to deliver *k* packets is the maximum routing time $T_{max}$ of $\left\{T_{1},T_{2},\cdots ,T_{n}\right\}$.

Pros and Cons. With end-to-end routing time as the metric, LCMR avoids heavily congested routes during its route discovery process. The total transmission time is also reduced by adjusting the number of packets assigned to a path according to the times that the least common multiple *L* has over the routing time $T_{i}$ of the path. However, evenly distributing traffic loads among different paths in the time domain can lead the path with the least end-to-end routing time to take the most traffic burden and can cause early node death. Also LCMR does not adapt well to network dynamics such as congestion during data transmission or route breakage. And the inability to guarantee path disjointness decreases the possible gain of multipath routing.

### 4.6. Conclusion and Future Trends

Multipath routing protocols play an important role in the QoS provision and multimedia data delivery in WMSNs. Interestingly, most multipath routing protocols for WMSNs have their single path counterparts in traditional WSNs. DGR and GEAM are based on GPSR [122]. ARA ant-colony-based routing [124] is the basis of AntSensNet. Z-MHTR is an extension of the Zigbee cluster-tree routing [125]. LCMR is built on AODV [123]. Compared to their single path counterparts, the multipath routing protocols are able to distribute the bulk multimedia data traffic more evenly among sensors and thus improving load balancing and fair energy consumption. However, for the routing protocols to obtain maximum gain from multiple concurrent paths, designs like [49,50] with inter-path interference awareness are favored in order to solve the route coupling problem [126]. Nonetheless, most multipath routing protocols put there emphasize solely on load balancing and energy conservation. For example, only AntSensNet and LCMR explicitly considers QoS and/or prioritization and service differentiation while, in fact, QoS should be a necessity of routing in WMSNs. Moreover, none of the multipath protocols consider about the nature of camera sensors in terms of directional sensor coverage and correlation within the multimedia data from neighboring nodes, which are important to reduce data redundancy and to improve quality of multimedia services [49,50,126].

We argue that the multipath routing protocol that will best fit the demands of WMSNs needs to be energy efficient, QoS-aware, and interference-aware. In the meantime, it also should consider the existence of heterogeneous traffics and correspondingly provide prioritization and differentiated services. Last but not the least, congestion control mechanisms, route recovery methods, and consideration of the nature of camera sensors are also merits in order to provide steady and quality multimedia services.

## 5. Conclusions

WMSNs are the driving force behind many multimedia applications in the age of Internet of things (IoT) thanks to their ability to produce multimedia surveillance data including image, video, and streaming media. Over the years, the research community has made much progress toward the proliferation of WMSNs. Nonetheless, challenges still exist due to the resource constraints and the unique characteristics of WMSNs. In this paper, we not only outline the characteristics of WMSNs, but at the same time we identify the respectively enforced requirements and for each specific requirement we sum up the existing solution approaches. In doing so, we also formally introduce many definitions of the requirements and solution approaches, e.g., intrinsic camera calibration and extrinsic camera calibration. Our work is also the first to provide an all-including big picture of WMSNs by presenting it as an inter-connected unibody. We believe such an illustration will act as the bond among policy makers, network planners, and researchers and therefore foster interdisciplinary cooperations and accelerate the advance of WMSNs. Moreover, we survey MAC protocols and routing protocols which are two major classes of communication protocols for data communication and QoS provision. Specifically, energy efficient MAC protocols with prioritization and service differentiation properties are our focus. We find that hybrid protocols of CSMA/CA and TDMA might hold the key for MAC in WMSNs, since they enable switching in between CSMA/CA for lower data rate communication to avoid channel under utilization and TDMA for high data rate communication to reduce collision and improve throughput. In terms of routing, we center around disjoint multipath routing protocols. To disseminate large amount of multimedia data across the network, we believe disjoint and interference aware multipath routing protocol is necessary as it is able to realize higher aggregated bandwidth with multiple concurrent paths. Although the existing multipath routing protocols mainly focus on load balancing and energy efficiency while most lack when it comes to QoS provisions. In the meantime, it is worth pointing out that under current literature, both MAC protocols and multipath routing protocols largely fail to consider the existence of camera sensors and the nature of multimedia data, which we argue should be the future focus of designing communication protocols for WMSNs.

## Figures and Tables

**Figure 1 sensors-19-00199-f001:**
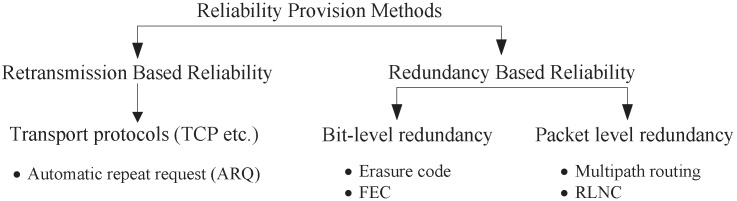
Reliability Provision Methods in wireless multimedia sensor networks (WMSNs).

**Figure 2 sensors-19-00199-f002:**
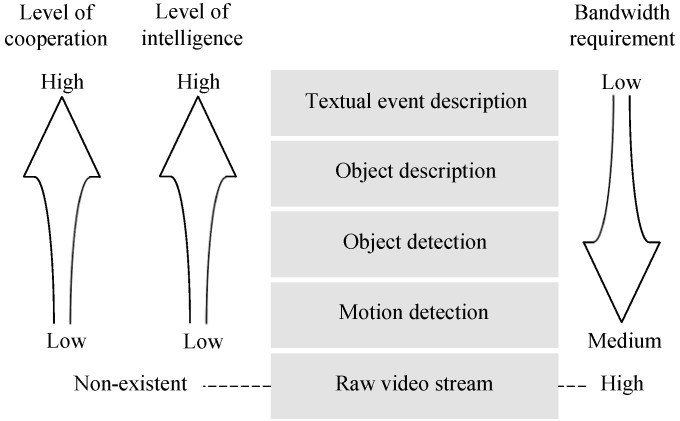
Local Processing in WMSNs.

**Figure 3 sensors-19-00199-f003:**
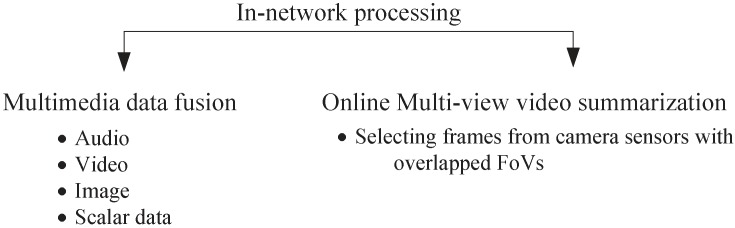
In-network Processing in WMSNs.

**Figure 4 sensors-19-00199-f004:**
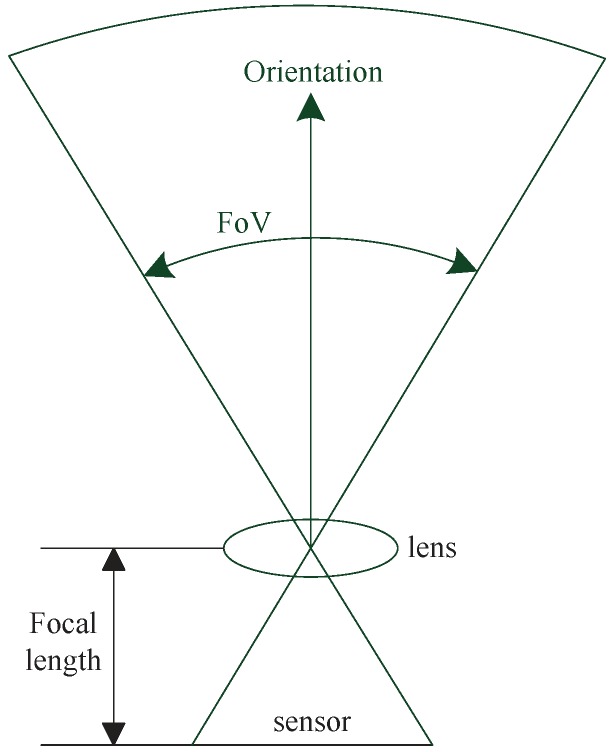
Field of View of A Camera Sensor.

**Figure 5 sensors-19-00199-f005:**
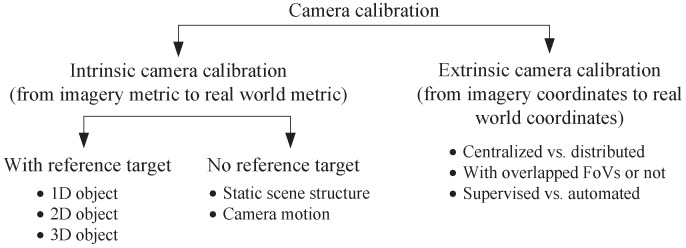
Camera Calibration in WMSNs.

**Figure 6 sensors-19-00199-f006:**
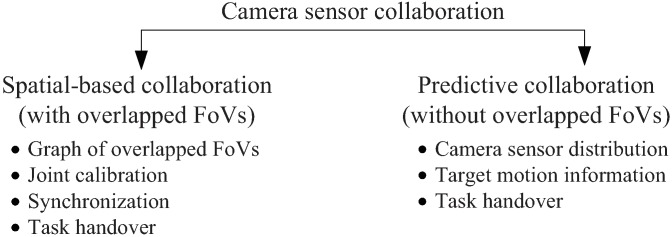
Camera Sensor Collaboration Schemes in WMSNs.

**Figure 7 sensors-19-00199-f007:**
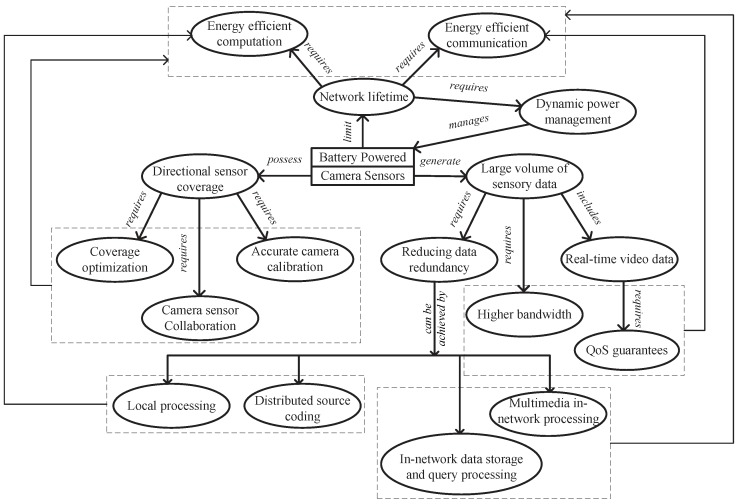
Characteristics and Requirements of WMSNs Presented as Unibody.

**Figure 8 sensors-19-00199-f008:**
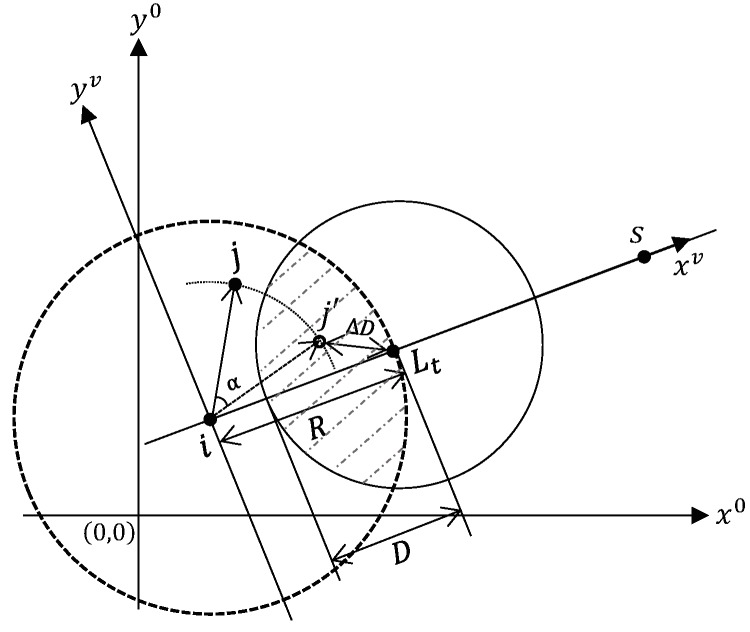
Next Hop Selection in DGR.

**Figure 9 sensors-19-00199-f009:**
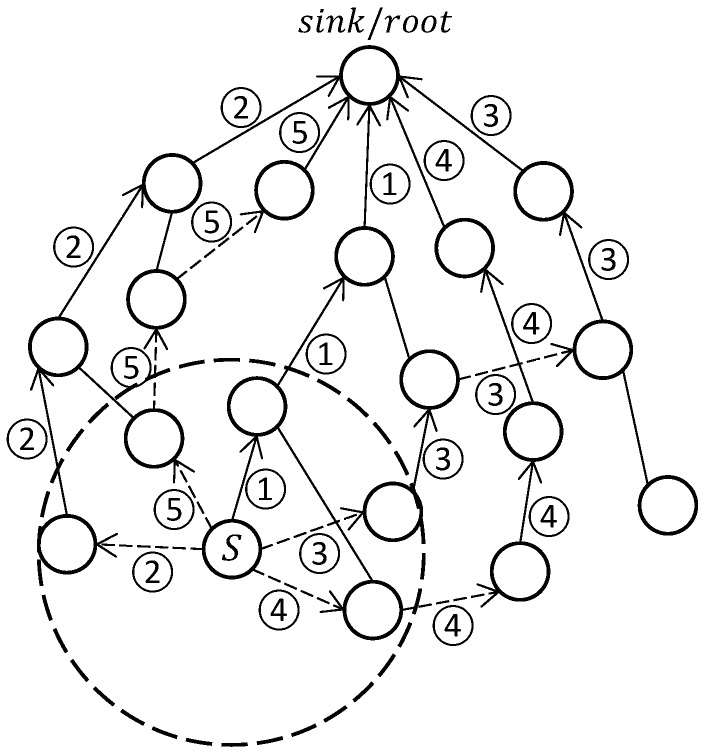
An Example of Z-MHTR.

**Figure 10 sensors-19-00199-f010:**
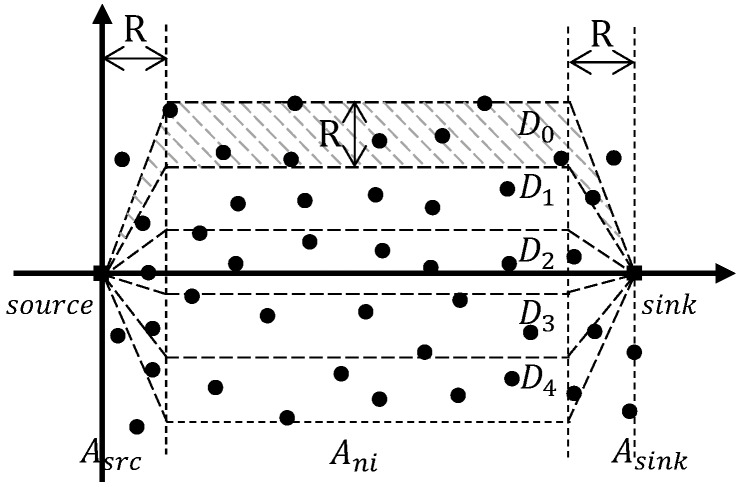
Division of Topology in GEAM.

**Table 1 sensors-19-00199-t001:** Characteristics and Requirements of wireless multimedia sensor networks (WMSNs).

Characteristics	Requirements	Approaches
Battery-powered	Energy efficiency	Energy-efficient computation
		• Image compression algorithms: [26,27,28,29,30,31]
		• Video compression algorithms: [32,33,34]
		Dynamic power management: [35,36,37]
		Energy-efficient communication
		• Transport protocols: [38]
		• MAC protocols: [39,40,41,42,43,44,45]
		• Routing protocols: [46,47,48,49,50,51]
Real-time video data	QoS guarantees	Delay guarantee
		• MAC protocols: [39,40,41,42,43,44,45]
		• Routing protocols: [47,51]
		Reliability guarantee: [13,52,53]
		• MAC protocols: [41,42,44]
		• Routing protocols: [46,47]
		Prioritization and service differentiation: [4,54]
		• MAC protocols: [39,40,41,42,43,44,45]
		• Routing protocols: [47]
	Quality of Experience	[55,56,57,58]
Large volume of multimedia data	Reducing data redundancy	Local processing: [59,60,61,62]
		Multimedia in-network processing
		• Multimedia data fusion: [63,64]
		• Multi-view video summarization [65]
		Distributed source coding: [27,30,34,66,67,68]
		In-network data storage and query processing: [69,70,71,72,73]
	Higher bandwidth requirement	Multi-channel MAC protocols: [74,75,76]
		Multipath routing: [46,47,48,49,50,51]
		Ultra Wideband technique: [77,78]
Directional sensor coverage	Accurate camera calibration	Intrinsic camera calibration: [79,80,81,82,83,84,85,86]
		Extrinsic camera calibration: [79,80,87,88,89,90,91,92,93,94]
	Coverage optimization	[95,96,97,98,99,100,101,102]
	Camera sensor collaboration	[103,104]

**Table 2 sensors-19-00199-t002:** A comparison of surveyed MAC protocols.

Protocols	EQ-MAC	Saxena et al. [40]	Diff-MAC	MQ-MAC	IH-MAC	AMPH	PA-MAC
**MAC mechanism**	hybrid of CSMA/CA and TDMA	CSMA/CA	CSMA/CA	IEEE 802.15.4	hybrid of CSMA/CA and TDMA	hybrid of CSMA/CA and TDMA	IEEE 802.15.4
**Synchronization**	global, precise	not required	not required	local, precise	local, loose	global, precise	global, precise
**QoS guarantee**	delay	throughput, delay	reliability, delay	reliability, delay	delay	reliability, delay	throughput, delay
**Prioritization schemes**	traffic types	traffic types	traffic types, traversed hop count of packets	traffic types, packet lifetime	traffic types	traffic types, dynamic	traffic types
**Service differentiation schemes**	dynamic slot allocation	adaptive contention window, dynamic duty cycle	adaptive contention window, dynamic duty cycle, weighted fair queuing	dynamic channel allocation, dynamic slot allocation, adaptive contention window	adaptive contention window, dynamic slot allocation	adaptive contention window, dynamic slot allocation	dynamic channel access time control
**Scalability**	poor	good	good	medium	medium	poor	poor
**Adaptation to dynamic traffics**	good	medium	medium	poor	good	good	poor
**Collision rate**	low	medium	medium	low	low	low	high
**Fairness**	poor/yes	medium/no	good/no	medium/no	medium/no	good/no	medium/no
**Energy efficiency**	good	medium	medium	good	medium	poor	good
**Message passing**	no	no	yes	no	no	yes	no
**Clustered**	yes	no	no	yes	yes	no	no
**Year**	2008	2008	2011	2015	2013	2014	2016

**Table 3 sensors-19-00199-t003:** A comparison of the surveyed multipath routing protocols.

Protocols	DGR	AntSensNet	Z-MHTR & Z. Bidai et al. [49]	GEAM	LCMR
**Routing method**	geographic routing	ant colony based routing	ZigBee cluster-tree routing	geographic rouing	ad-hoc on-demand distance vector routing
**Routing metric**	geographic distance and deviation angle	pheromone value of residual energy, delay, packet loss rate and available memory	network address	geographic distance	end-to-end delay
**Routing states**	one hop neighbor table	one hop neighbor table, routing pheromone table	one hop neighbor table, branches used for tree routing, and/or interfering node table	one hop neighbor table, district information	routing table
**Disjoint paths**	yes	yes	yes	yes	no
**QoS metrics**	reliability, throughput	reliability, delay, throughput	throughput	throughput	delay, throughput
**Path repair**	yes	yes	no	yes	no
**Scalability**	good	good	good	good	poor
**Congestion control**	no	yes	no	no	no
**Prioritization**	no	yes	no	no	no
**Service differentiation**	no	yes	no	no	no
**Energy efficiency**	medium	medium	good	good	poor
**Clustered**	no	yes	yes	no	no
**Interference aware**	no	no	yes	yes	no
**Year**	2007	2010	2012 & 2014	2013	2017

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
