# Peer review of "A Survey of Energy-Efficient Communication Protocols with QoS Guarantees in Wireless Multimedia Sensor Networks"

_sensors, 2019, doi:10.3390/s19010199_

Reviewer 1 Report

The authors provide a good high-level overview for QoS in the context of WMSNs, with detail evaluations for MAC and Network layer QoS considerations. The paper is relevant and provides a good overview for practitioners. A little bit of proofreading might benefit the paper (leaning to a minor revision). I would suggest to include a short narrative on the actual content generation part to make the overall survey holistic, i.e., include one or two pages overall that deal with the media encoding and media characteristics in the context of WMSNs, just as for the MAC/Routing components evaluated. That would round out the idea of QoS, as no matter how the sensors are organized and stream, if content coding provides low-level QoS (say, image quality), any lower layer enhancements are not likely improving the overall QoS. This last point would result in a slightly larger revision, hence the rating. 

Author Response

Thanks for your time to review our work and your kind suggestions.

In response to your advices, we have prepared our answers.

You may find the our answer in the attached PDF file.

Regards.

Reviewer 2 Report

There are several papers missing from the survey such as the following

Correlation-Aware QoS Routing With Differential Coding for Wireless Video Sensor Networks

Wireless multimedia sensor network technology: A survey

QoS and Energy-Aware Dynamic Routing in Wireless Multimedia Sensor Networks

Energy Efficiency QoS Assurance Routing in Wireless Multimedia Sensor Networks

Author Response

(The authors gave the same response as above.)

Reviewer 3 Report

It is a review article on the state of the art in Wireless Multimedia Sensor Networks. A few state-of-the-art reviews on this topic have been published in recent years, most of which are cited in this paper, one of them in the Sensors journal in 2010. This review focuses mainly on describing the main challenges, and in terms of solutions, it focuses on MAC levels and routing.

The strong points of the paper is that it is very well written and structured, being easy to read and clear. The review of the literature is very well structured, dividing it into categories, with their pros and cons and summarizing the main conclusions at the end of each section.

The weak points are:

-        Although it is appreciated how well structured it is, it is not too new because other reviews of the state of the art of the same subject have already been published.

-        Section 2 discusses many aspects related to energy consumption and the use of cameras in the nodes of the WMSN, which are not covered later in the revision of the following sections.

-        Related to the previous point, gives the impression that all the nodes in the WMSN will be equipped with a camera (ability to generate multimedia content) when in a real network it is expected a few nodes producing multimedia content, other to be just consumers of multimedia content, and others only routers of that traffic.

-        There are few references from the last two years. I have found some that may be relevant (this is just a suggestions, none of the following authors has relation with this reviewer):

o   M. Z. Hasan, H. Al-Rizzo and F. Al-Turjman, "A Survey on Multipath Routing Protocols for QoS Assurances in Real-Time Wireless Multimedia Sensor Networks," in IEEE Communications Surveys & Tutorials, vol. 19, no. 3, pp. 1424-1456, thirdquarter 2017. doi: 10.1109/COMST.2017.2661201

o   Tarek AlSkaif, Boris Bellalta, Manel Guerrero Zapata, Jose M. Barcelo Ordinas, "Energy efficiency of MAC protocols in low data rate wireless multimedia sensor networks: A comparative study, " Ad Hoc Networks, Volume 56, 2017, Pages 141-157, ISSN 1570-8705, https://doi.org/10.1016/j.adhoc.2016.12.005.

o   M. Usman, N. Yang, M. A. Jan, X. He, M. Xu and K. Lam, "A Joint Framework for QoS and QoE for Video Transmission over Wireless Multimedia Sensor Networks," in IEEE Transactions on Mobile Computing, vol. 17, no. 4, pp. 746-759, 1 April 2018. doi: 10.1109/TMC.2017.2739744

o   F. Al-Turjman and A. Radwan, "Data Delivery in Wireless Multimedia Sensor Networks: Challenging and Defying in the IoT Era," in IEEE Wireless Communications, vol. 24, no. 5, pp. 126-131, October 2017. doi: 10.1109/WCM.2017.1700054

o   Al-Ariki, H.D.E. & Swamy, M.N.S. Wireless Netw (2017) 23: 1823. https://doi.org/10.1007/s11276-016-1256-5

Overall, my assessment of the paper is positive.

Author Response

Thanks for your time to review our work and your kind suggestions.

In response to your advices, we have prepared our answers.

You may find the our answer in the attached PDF file.

Regards.

Round  2

Reviewer 1 Report

The authors have added additional references that were requested by this reviewer as well as an additional QoE discussion.

Reviewer 2 Report

All the recommendations/ concerns are properly addressed